# Comparison of Impressions of COVID-19 Vaccination and Influenza Vaccination in Japan by Analyzing Social Media Using Text Mining

**DOI:** 10.3390/vaccines11081327

**Published:** 2023-08-05

**Authors:** Yoshiro Mori, Nobuyuki Miyatake, Hiromi Suzuki, Yuka Mori, Setsuo Okada, Kiyotaka Tanimoto

**Affiliations:** 1Department of Hygiene, Faculty of Medicine, Kagawa University, Miki 761-0793, Japan; miyatake.nobuyuki@kagawa-u.ac.jp (N.M.); suzuki.hiromi@kagawa-u.ac.jp (H.S.); 2Sakaide City Hospital, Sakaide 762-8550, Japan; hosp02@city.sakaide.lg.jp (S.O.); taka12ki05@gmail.com (K.T.); 3Institute of Biomedical Sciences, Tokushima University Graduate School, Tokushima 770-8503, Japan; mori.yuka@tokushima-u.ac.jp

**Keywords:** Twitter^®^, social media, text mining, COVID-19, influenza

## Abstract

The aim of this study was to compare impressions of COVID-19 vaccination and influenza vaccination in Japan by analyzing social media (Twitter^®^) using a text-mining method. We obtained 10,000 tweets using the keywords “corona vaccine” and “influenza vaccine” on 15 December 2022 and 19 February 2023. We then counted the number of times the words were used and listed frequency of these words by a text-mining method called KH Coder. We also investigated concepts in the data using groups of words that often appeared together or groups of documents that contained the same words using multi-dimensional scaling (MDS). “Death” in relation to corona vaccine and “severe disease” for influenza vaccine were frequently used on 15 December 2022. The number of times the word “death” was used decreased, “after effect” was newly recognized for corona vaccine, and “severe disease” was not used in relation to influenza vaccine. Through this comprehensive analysis of social media data, we observed distinct variations in public perceptions of corona vaccination and influenza vaccination in Japan. These findings provide valuable insights for public health authorities and policymakers to better understand public sentiment and tailor their communication strategies accordingly.

## 1. Introduction

The first case of coronavirus infectious disease, also known as COVID-19, originated in 2019 and was reported in China on 8 December 2019 [1]. Subsequently, on 16 January 2020, the first case of COVID-19 in Japan was confirmed. Since then, the spread of COVID-19 infection has continued to affect Japan [2]. As of 26 February 2023, the cumulative number of novel positive cases of COVID-19 in Japan reached a staggering 33,184,966 [3].

In an ongoing effort to control the spread of the virus, the Japanese government has been actively encouraging its citizens to modify their behavior while also initiating a comprehensive COVID-19 vaccination campaign in February 2021 [4]. As of 23 February 2023, the vaccination coverage for COVID-19 among the entire population stood at 77.9% for the first dose, 77.4% for the second dose, 68.3% for the third dose, 45.9% for the fourth dose, and 23.7% for the fifth dose. Unfortunately, the vaccination rate after the third dose has shown limited progress [5]. Additionally, reports indicate that the vaccination rate remains lower among younger generations compared to older generations [6].

While influenza infection poses a significant public health challenge globally, Japan has not experienced any major influenza epidemics since the emergence of COVID-19 [7]. However, the influenza vaccination rates for the 2021–2022 period were considerably lower compared to pre-pandemic levels [8]. An influenza epidemic was anticipated for the 2022–2023 season, and the number of confirmed influenza cases had been steadily increasing since November 2022. From 13 February 2023 to 4 March, a total of 62,101 cases were reported, marking a substantial increase of approximately 2400-fold compared to the same period in the previous year [9].

Social media platforms have revolutionized communication, enabling individuals to share their thoughts, opinions, and ideas on a global scale. In Japan, these platforms have gained immense popularity, transcending generational boundaries. The Ministry of Internal Affairs and Communications conducted a survey in 2021, revealing that LINE^®^ remains the dominant platform with a staggering 92.5% usage rate, followed by Facebook at 32.6%. Instagram^®^ and Twitter^®^ have also gained considerable traction, with respective usage rates of 48.5% and 46.2% [10]. Twitter^®^, known for its succinct 140-character messages [11], facilitates connections between users who share similar interests and concerns. With its viral nature, information spreads rapidly through retweets, making it an invaluable tool for disseminating important updates. What sets Twitter^®^ apart is its diverse user base, spanning across different age groups. Astonishingly, 67.4% of teenagers, 78.6% of people in their 20s, 57.9% of individuals in their 30s, 44.8% of those in their 40s, 34.3% of people in their 50s, and even 14.1% of individuals in their 60s utilize Twitter^®^ in Japan [10].

Social media use continues to grow worldwide, and there are numerous tweets about various health problems, including human papillomavirus infection [12], suicide [13], stress [14], air pollution [15], and others [12,13,14,15,16,17,18,19,20]. Keelan et al. examined 303 posts on HPV immunization and classified 157 (52%) as positive, 129 (43%) as negative, and 17 (6%) as ambivalent [12]. Another study investigated the relationship between suicide-related tweets and suicidal behavior in order to identify young individuals at risk on the Internet [13]. Song et al. obtained social data from online news and blogs, which provided insights into adolescent stress [14]. Additionally, a study collected air pollution data from 2210 monitoring sites across China, used keyword-based filtering to identify individual messages related to air pollution and health on Chinese Twitter^®^, and revealed the correlation between air pollution and related tweets during periods and locations with poor air quality [15]. Therefore, analyzing data from social media can be beneficial for examining health concerns within communities.

Previous studies have examined COVID-19 by analyzing social media [21,22,23,24,25]. Shim et al. utilized Google search trends to investigate the impact of media on skin lesions during the COVID-19 pandemic [21]. Abd-Alrazaq et al. extracted text and metadata (user profile information including the number of likes, retweets, and number of followers) from Twitter^®^, revealing that citizens were concerned about economic losses during the COVID-19 epidemic [22]. Pobiruchin et al. also analyzed Twitter^®^ based on 16 hashtags about COVID-19 in the European region to examine temporal changes and regional characteristics of COVID-19-related tweets [23]. Alhuzali et al. extracted over 500,000 tweets related to COVID-19 from 48 different cities in the U.K. between February 2020 and November 2021, initially observing positive opinions that later shifted towards an increase in negative opinions [24]. In Japan, Suzuki et al. used Twitter^®^ to examine the Japanese public’s attitude towards discrimination related to COVID-19 [25].

Understanding the extensive reach and popularity of Twitter^®^, it becomes clear that analyzing Twitter^®^ data can provide invaluable insights, particularly in promoting COVID-19 vaccination among younger generations. By closely examining the sentiments, concerns, and preferences expressed by Twitter^®^ users, effective strategies can be developed to address any doubts or misconceptions related to vaccinations. Engaging with the younger audience through Twitter^®^ can play a pivotal role in significantly increasing vaccination rates in Japan and effectively combatting the ongoing pandemic. Previously, we reported on the initial impressions of medical staff regarding the COVID-19 vaccine using text mining, obtaining text data through a self-reported questionnaire [26,27]. In younger women, the words “pregnancy” and “side effect” were frequently associated with the first vaccine, while the mention of “side effect” decreased for the second vaccine [26]. Additionally, “pregnancy” was frequently mentioned in tweets about the third vaccine by women in their 30s [27]. However, the text data were obtained only from a limited number of medical staff at one hospital in Japan.

Despite the potential usefulness of the analysis of social media, particularly platforms like Twitter^®^, there has been a significant lack of comparative studies examining public sentiments regarding COVID-19 vaccination and influenza vaccination in Japan. This knowledge gap poses a hindrance to our comprehension of public perceptions and presents a formidable obstacle to the advancement of future COVID-19 vaccination initiatives. In order to bridge this gap and make valuable contributions to the improvement of COVID-19 vaccination efforts, we embarked on a comprehensive text-mining analysis of Twitter^®^ data. Our objective was to delve into and draw comparisons between the general impressions held by the public concerning COVID-19 vaccination and influenza vaccination in Japan.

## 2. Materials and Methods

### 2.1. Data Extraction and Adjustment

To gather data for our study, we made use of a tool called the Twitter^®^ application programming interface (API) [28]. The API is a set of rules and protocols that allows us to access information from Twitter^®^, such as tweets posted by users. Specifically, we used the API to retrieve tweets that were posted within the last seven days. We started the process of extracting data from Twitter^®^ on 15 December 2022, which was before the widespread occurrence of influenza infections [9]. To narrow down our search and focus on the topic of vaccinations related to both the coronavirus and influenza viruses, we used two specific keywords: “corona vaccine” and “influenza vaccine”. Initially, we retrieved 10,000 tweets that were relevant to discussions about vaccinations in the context of both viruses. The purpose of this initial extraction was to capture a wide range of tweets related to our topic of interest. Later, on 19 February 2023, when the influenza virus was more prevalent [9], we performed an additional extraction of 10,000 tweets from Twitter^®^ using the same set of keywords as before. This allowed us to gather more recent data specifically related to discussions about vaccinations during the time when the influenza virus was actively circulating.

To carry out the extraction process, we used a programming language called R^®^ [29], along with an integrated development environment called RStudio^®^ [30]. R^®^ is a popular programming language used for analyzing data, and it provides a wide range of features for manipulating data and performing statistical analysis. On the other hand, RStudio^®^ is a specialized development environment created specifically for working with R^®^. It offers a user-friendly interface and various tools that make coding and analysis tasks easier to perform. By utilizing R^®^ and RStudio^®^ together, we were able to efficiently work with data and conduct statistical analysis in a convenient and effective manner.

To collect the tweets, we used programming code in R^®^ and RStudio^®^. We set specific keywords, namely “corona vaccine” and “influenza vaccine”, to filter the tweets. The extraction process was designed to include instances where the keywords appeared separately within a tweet. For example, a tweet could contain “corona” and “vaccine” in different parts of the text or “influenza” and “vaccine” in separate sections.

After the initial extraction, we performed further adjustments on the extracted data using Visual Studio Code^®^ [31]. Visual Studio Code^®^ is a popular source code editor that offers a range of features and extensions to support various programming languages, including R^®^. In our case, we utilized Visual Studio Code^®^ to perform data cleaning and formatting tasks. During the adjustment phase, we removed certain elements from the extracted data to ensure data privacy and integrity. These elements included the following:Uniform resource locators (URLs): Any web addresses present in the tweets were eliminated, as they were not relevant to our analysis;Twitter^®^ account names: The usernames associated with each tweet were removed to protect the privacy of the individuals who posted them;Blank lines: Empty lines within the data were eliminated, as they did not contribute to the analysis;Phone numbers and zip codes: Any phone numbers and zip codes mentioned in the tweets were removed to maintain privacy and comply with data-protection regulations;Advertisements and suspected advertisements: Tweets that appeared to be advertisements or promotional in nature were excluded from the dataset, as our focus was on genuine discussions and opinions related to vaccination.

By performing these adjustments, we ensured that the data used in our subsequent analysis was clean, reliable, and devoid of any identifiable or irrelevant information.

### 2.2. Text Mining 

Following the data extraction and adjustment process, we carefully conducted a comprehensive analysis of the tweet data using a powerful and widely recognized text-mining method called KH Coder (KH Coder 3.0, Koichi Higuchi, Tokyo, Japan). This methodology has been previously reported [32,33] and is highly regarded for its ability to extract valuable insights from textual data.

We used KH Coder to carefully analyze the tweets we collected about the “corona vaccine” and “influenza vaccine”. Our analysis involved both quantitative and qualitative techniques to thoroughly understand the patterns, recurring themes, and complex relationships in the tweets. To start, we created a detailed frequency table to identify the most commonly used words in the dataset. This information helped us gain a better understanding of the main topics and ongoing discussions related to the vaccines.

In addition to frequency analysis, we took advantage of the advanced multivariate analysis capabilities offered by KH Coder. This allowed us to dive deeper into the tweet data and extract more nuanced insights. To visually represent the relationships between the words, we used a popular technique called multi-dimensional scaling (MDS). MDS arranges the tweets as points in a low-dimensional space, highlighting similarities and differences between them. This MDS plot provided a clear visual representation of how tweets were related, with similar words positioned close together and dissimilar words positioned farther apart. This approach has been successfully used in previous studies to uncover distinct clusters and intricate patterns in text data [34,35]. Cluster analysis was then performed based on the coordinates of the plots obtained by the MDS method. The method is based on the Ward method using Euclidean distance. In this case, words that are placed close to each other on the plot are classified into the same cluster. In this case, eight clusters, which is the default, were used to support the interpretation of the results of the MDS method.

By using these strong analytical methods and techniques, we gained deep understanding from the tweet data. We discovered important discussions, uncovered new patterns and connections, and learned valuable information about the “corona vaccine” and “influenza vaccine” trends in social media conversations. This analysis is a valuable tool for better comprehending the dynamics and trends surrounding these topics. 

### 2.3. Ethics 

Ethical approval for this study was obtained from the Ethical Committee of Sakaide City Hospital in Sakaide, Japan (Number: 2022-002, Date: 15 December 2022). The committee carefully reviewed and approved the research methodology, ensuring that the study adhered to ethical guidelines and considerations for the protection of participants’ rights and privacy and the responsible use of social media data.

### 2.4. English Editing 

The original text of this paper was written by the authors. However, English editing was performed by a professional proofreader (Medical English Service, Kyoto, Japan). Chat GPT and DeepL translation software were also used for some of the post-review English editing.

## 3. Results

On 15 December 2022, a study was conducted to analyze tweets related to the “corona vaccine” and the “influenza vaccine”. Among the 10,000 tweets examined, the phrase “corona vaccine” was used 552,432 times, while “influenza vaccine” was used 576,708 times. Table 1 and Table 2 provide lists of frequently used words among nouns and adjectival nouns extracted from these tweets. In the tweets discussing the “corona vaccine”, the most commonly used word was “vaccination”, which appeared 7625 times. Other frequently used words included “vaccine”, “corona vaccine”, “corona”, “new type”, and “death”. On the other hand, in tweets about the “influenza vaccine”, the word “vaccine” appeared most frequently, with a count of 8429. This was followed by “vaccination”, “influenza vaccine”, “influenza”, “corona”, and “severe disease” (Table 1).

Moving forward to 19 February 2023, the number of tweets mentioning the “corona vaccine” increased to 600,591, while the “influenza vaccine” was mentioned 577,965 times. Among the tweets focusing on the “corona vaccine”, the most commonly used word remained “vaccine”, with a count of 7811. It was followed by “corona vaccine”, “vaccination”, “corona vaccine”, “new type”, and “after effect”. In the case of tweets discussing the “influenza vaccine”, “vaccine” was once again the most frequently used word, appearing 9411 times. Other notable words included “influenza”, “mask”, “WHO”, “world”, and “vaccination” (Table 2).

The collected tweets were subjected to analysis using multi-dimensional scaling (MDS) to identify clusters of related words or concepts. In the analysis conducted in December 2022 (Figure 1), it was observed that tweets mentioning “corona vaccine” formed clusters associated with the word “death”. Additionally, a separate cluster emerged regarding the cost of future vaccinations. In the case of “influenza vaccine” tweets, a cluster was formed around the term “severe disease”, which was distinct from the cluster related to the “influenza vaccine” itself.

Moving on to the analysis performed in February 2023, the “corona vaccine” tweets revealed a separate cluster associated with the term “after effect”, which was distinct from the main cluster representing the “corona vaccine”. This cluster was related to concepts such as the “Ministry of Health, Labor and Welfare”, “medical care”, and “support”. Notably, the usage of “severe disease” was not observed in these tweets, and overall, there were no significant changes compared to the December 2022 analysis (Figure 2).

In summary, the analysis of tweets revealed shifts in the frequency and usage of words related to the “corona vaccine” and the “influenza vaccine” over time. These findings provide insights into the public discourse surrounding these vaccines and the evolving concerns and topics associated with them.

## 4. Discussion

In the present study, we compared public impressions of corona vaccination and influenza vaccination in Japan by analyzing Twitter^®^ using a text-mining method for the first time. Prior to the influenza pandemic (on 15 December 2022), the impression of younger generations of the corona vaccine was characterized by the word “death”, while that of the influenza vaccine was associated with “severe disease”. During the influenza pandemic (on 19 February 2023), the use of the word “death” decreased, and a new association with “after effect” was identified in relation to the corona vaccine, while the word “severe disease” was no longer detected in tweets discussing the influenza vaccine.

According to impressions of the corona vaccine based on an analysis of social media, Awijen et al. reported an increase in Google search trends related to fear and anxiety with the spread of corona vaccination [36]. In a text-mining study on data related to the corona vaccine from Twitter^®^, Lyu et al. analyzed 1,499,421 tweets containing the words “vaccine”, “immunization”, and “vaccination”. An emotion analysis revealed that trust was the most predominant emotion, followed by anticipation, fear, and sadness [37]. Ruiz-Núñez examined the tweets of healthcare providers in Spain at two different time periods (one closest to the 2021 suspension of the AstraZeneca vaccine and the other 30 days later), reporting that negative tweets sent by healthcare providers against the corona vaccine had an impact on the prevention of COVID-19 [38]. Ong et al. investigated tweets from Malaysian Twitter^®^ users related to additional doses of vaccines and found that they were interested in the type of vaccine, the effect of the vaccine, and the vaccination schedule [39]. Previous studies also examined the impression of the influenza vaccine using social media [40,41]. Guidry et al. analyzed Twitter^®^ before and during an influenza pandemic to assess changes in attitudes towards influenza vaccination during the influenza season [40]. Signorini et al. used Twitter^®^ to examine disease outbreaks and public interest in the USA during an influenza pandemic, suggesting the potential of automated extraction of epidemiological information using social media [41].

In the present study, we conducted a groundbreaking analysis to investigate the contrasting public impressions of corona vaccination and influenza vaccination in Japan. Utilizing a text-mining method on Twitter^®^, we embarked on an unprecedented approach, marking a significant advancement in vaccine-perception research. The results of our study revealed a stark divergence between the two vaccines. To contextualize our findings, we referred to a previous survey conducted by Katsiroumpa et al., where 254 high-risk individuals were surveyed regarding influenza vaccination during the COVID-19 epidemic. Their findings indicated that 39.4% expressed a desire to be vaccinated, while 33.9% did not. Intriguingly, individuals who had received a booster dose of the corona vaccine demonstrated a higher inclination to be vaccinated against influenza. However, they also reported that side effects of the corona vaccine and fatigue diminished their willingness to receive the influenza vaccine [42]. Notably, this study was the first to directly compare public impressions of corona vaccination and influenza vaccination in Japan. By harnessing the power of social media analysis, including Twitter^®^, we were able to promptly access and analyze the opinions and impressions of the general public, with a particular focus on the younger generation. The disparities in public perception of the corona vaccine and influenza vaccine were found to be significant.

Moreover, we made an intriguing observation regarding the association of specific terms with the corona vaccine on Twitter^®^. On 15 December 2022, the term “death” was categorized alongside the “corona vaccine”. However, by 19 February 2023, a separate cluster had emerged, linking the “after effect” to the “corona vaccine”. This cluster was also associated with other terms such as “Ministry of Health, Labor and Welfare”, “medical care”, and “support”. These findings suggest that the term “after effect” was not directly linked to negative connotations associated with the “corona vaccine”. This insight highlights the value of Twitter^®^ data in informing future policy decisions regarding vaccination. Drawing inspiration from the work of Brownstein et al., who proposed a method for tracking disease outbreaks and public interest through social media analysis, we acknowledge the potential of social media platforms, particularly Twitter^®^, in enhancing healthcare policies [43]. In some of the literature, the importance of health strategies against COVID-19 has been shown [44,45]. Additionally, it has been reported that not only health strategies but also the importance of economic and global factors [46,47,48,49] are significant in preventing and improving COVID-19. In February 2023, the Ministry of Health, Labor, and Welfare in Japan released its future policies regarding corona vaccination [50]. In line with this, we firmly believe that continuous and systematic analysis of social media, including Twitter^®^, will be instrumental in raising public awareness about corona vaccination, addressing concerns and misinformation, and shaping future policies. 

However, we need to consider and address several limitations in our current study. Firstly, it is important to note that Twitter^®^ usage was most prevalent among individuals in their 20s, accounting for 78.6% of the sample, but only 46.2% across all age groups. As a result, the findings of our study may not equally represent information from all age demographics. Secondly, our study did not collect data on the clinical characteristics of participants, such as their gender, age, and occupation. This limitation restricts our ability to conduct more detailed and comprehensive analyses. Thirdly, Twitter^®^ is a platform where information is constantly updated in real time, and tweets are generated at an extremely fast pace, necessitating prompt data analysis. Real-time analysis requires fast and efficient data processing support. However, there was a certain amount of time required from data acquisition to analysis, interpretation, and the publication of the paper. Lastly, Twitter^®^ tweets have a character limit, and their brevity and use of text abbreviations can make analysis challenging. Furthermore, the wide range of expressions used in tweets, including slang, pictures, and hashtags, adds complexity to the analysis. To address this issue, we established certain rules in our study.

Despite the complexity, our study has shown that analyzing Twitter^®^ can greatly help promote corona vaccination in Japan. To encourage vaccination in the future, it is important to provide the younger generation with more detailed and carefully explained information about the potential effects of the corona vaccine. Additionally, this analysis can be beneficial for shaping future vaccine policies when dealing with unfamiliar infectious diseases. By understanding public opinions and feelings, we can develop strategies to enhance vaccine acceptance. This research sets the stage for future endeavors that utilize social media data for public health initiatives, ultimately resulting in improved healthcare policies and a better-informed population. 

## 5. Conclusions

In conclusion, our comprehensive analysis of Twitter^®^ using a text-mining method revealed distinct and noteworthy differences in the public’s impressions of corona vaccination and influenza vaccination. In Japan, the terms “death” and “after effect” were frequently associated with the corona vaccine, while “severe disease” was commonly mentioned in relation to the influenza vaccine, particularly among the younger generations. These findings show the public discourse surrounding vaccination and emphasize the importance of targeted communication strategies to address concerns and misconceptions.

## Figures and Tables

**Figure 1 vaccines-11-01327-f001:**
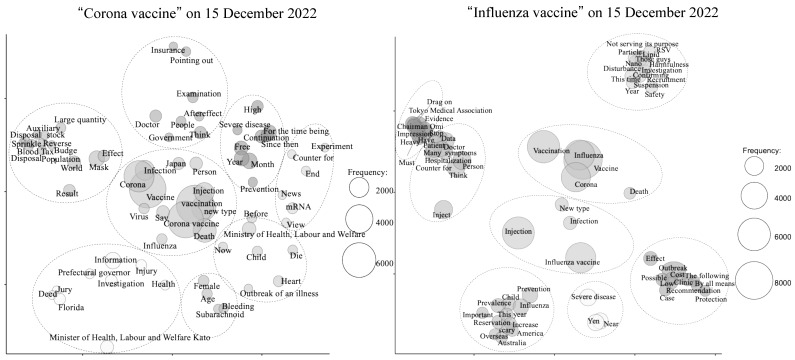
Concepts in data based on groups of words that often appear together or groups of documents that contain the same words on 15 December 2022 using multi-dimensional scaling (MDS).

**Figure 2 vaccines-11-01327-f002:**
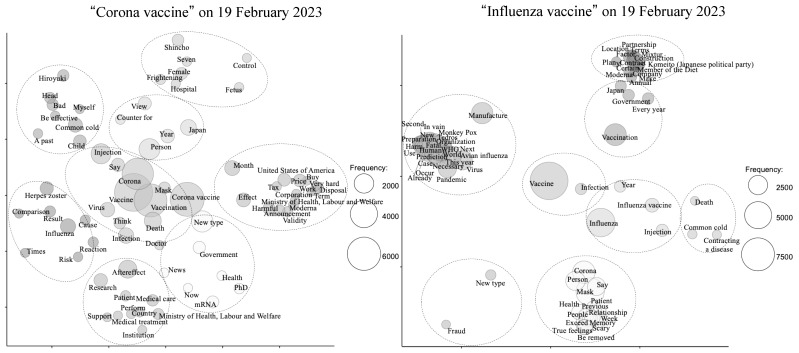
Concepts in data based on groups of words that often appear together or groups of documents that contain the same words on 19 February 2023 using multi-dimensional scaling (MDS).

**Table 1 vaccines-11-01327-t001:** List of frequently used words among nouns and adjectival nouns in a survey performed on 15 December 2022.

	Corona Vaccine	Times		Influenza Vaccine	Times	
		Used			Used	
	Total	552,432	%	Total	576,708	%
1st	Vaccination	7625	1.38	Vaccine	8429	1.46
2nd	Vaccine	7226	1.31	Vaccination	6049	1.05
3rd	Corona vaccine	6835	1.24	Influenza vaccine	5211	0.90
4th	Corona	4943	0.89	Influenza	5108	0.89
5th	New type	3161	0.57	Corona	4916	0.85
6th	Death	2855	0.52	Severe disease	1620	0.28
7th	Infection	1999	0.36	Doctor	1549	0.27
8th	Information	1331	0.24	Outbreak	1486	0.26
9th	Mask	1255	0.23	Hospitalization	1406	0.24
10th	Ministry of Health, Labor, and Welfare	981	0.18	Flu (influenza)	1391	0.24
11th	Free	981	0.18	Prevention	1369	0.24
12th	For the time being	901	0.16	Symptoms	1363	0.24
13th	World	897	0.16	Data	1202	0.21
14th	Continuation	872	0.16	Patients	1174	0.20
15th	Minister of Health, Labor, and Welfare Kato	859	0.16	Evidence	1155	0.20
16th	Since then	844	0.15	Impression	1137	0.20
17th	Japan	757	0.14	Epidemic	1136	0.20
18th	Doctor	739	0.13	Chairman Omi	1132	0.20
19th	Florida	721	0.13	Tokyo Medical Association	1131	0.20
20th	Investigation	716	0.13	Effect	1096	0.19

**Table 2 vaccines-11-01327-t002:** List of frequently used words among nouns and adjectival nouns in a survey performed on 19 February 2023.

	Corona Vaccine	Times		Influenza Vaccine	Times	
		Used			Used	
	Total	600,591	%	Total	577,965	%
1st	Vaccine	7811	1.30	Vaccine	9411	1.63
2nd	Corona vaccine	6248	1.04	Influenza	5682	0.98
3rd	Vaccination	5944	0.99	Mask	5412	0.94
4th	Corona vaccine	5386	0.90	WHO	3981	0.69
5th	New type	2436	0.41	World	3784	0.65
6th	After effect	1756	0.29	Vaccination	3276	0.57
7th	Death	1502	0.25	Manufacture	3183	0.55
8th	Japan	1326	0.22	Corona	2963	0.51
9th	Infection	1290	0.21	Avian influenza	2421	0.42
10th	Influenza	1227	0.20	Virus	2414	0.42
11th	Virus	1104	0.18	Pandemic	2246	0.39
12th	Common cold	1070	0.18	Organization	1968	0.34
13th	Effect	1011	0.17	Tedros	1929	0.33
14th	Research	966	0.16	Human	1917	0.33
15th	Herpes zoster	942	0.16	Use	1896	0.33
16th	United States	857	0.14	Prediction	1877	0.32
17th	Buy	845	0.14	Preparation	1876	0.32
18th	Children	836	0.14	Monkey pox	1876	0.32
19th	Government	789	0.13	Fatal	1866	0.32
20th	mRNA	778	0.13	In vain	1866	0.32

## Data Availability

Not applicable.

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
