# Peer review of "Comparison of Impressions of COVID-19 Vaccination and Influenza Vaccination in Japan by Analyzing Social Media Using Text Mining"

_vaccines, 2023, doi:10.3390/vaccines11081327_

Round 1

Reviewer 1 Report

Public impressions of corona and influenza vaccinations were studies via twitter word mining. As the authors state, the results may be useful for vaccination policies and awareness in the future.

1. What is the main question addressed by the research?

-main question addressed ;If there is any difference in public awaraness before and after covid vaccination?

2. Do you consider the topic original or relevant in the field? Does it address a specific gap in the field?

-I consider that topic is original

3. What does it add to the subject area compared with other published material?

-It adds public knowledge on the subject using word mining from social media.

4. What specific improvements should the authors consider regarding the methodology? What further controls should be considered?

-Nothing  is needed

5. Are the conclusions consistent with the evidence and arguments presented and do they address the main question posed?

-Conclusions are greatly appreciated

6. Are the references appropriate?

-Yes

7. Please include any additional comments on the tables and figures.

-No comment

Reviewer 2 Report

This is an interesting manuscript that needs some further work in order to become a good published paper. The idea of the research, that is to compare Tweets about Corona vaccines and Influenza vaccines in Japan at two distinct points in time is a great one. Further, the metods of sampling and the analysis of the ensuing data is an interesting one and is well carried out. However, there is room for improvement of the manuscript and below are some suggestions that I want to atuhors to take into account for the final version of the manuscript.

1) Some parts of this manuscript need to be re-written by the authors.

a) In particular, Section 4 Discussion. In that section there are whole paragraphs that do not beloing there at all and should be moved into the second part of the introduction. For example, the text in lines 138-164 and lines 172-189. The Discussion section should only discuss the results of the study in the context of the litarature review presented in the section Introduction. 

b) I suggest that the authors put in more efforts to writing the conclusions, that section is seriously to thin at the moment. If the authors want their study to be a substantial input into the promotion of Corona vaccines in Japan in the furutre, they really need to say how they want to do that, as it it not at all clear in the text as it reads at the moment. 

2) I also have some point to make about Figures 1 and 2. I much appreciate the application of KH Coder as methodology but the presentation of the results in Figs 1 and 2 really needs some further work done

a) Can the authors please sort out the clarity of the font the use for the labels in the graphs? As it is now it is very difficult to read.

b) There is nothing in the text that explains how the authors came up with the dotted circles that surround the grouping of themes and that really needs to be explained.

c) And this is the most important point, I would like the authors to explain better in the text the differences between Figures 1 and 2 as that difference is important for the overall argument they are making. 

If the authors decide to follow these suggestions I am confident that the manuscript will be much improved. 

There are no major issues with the quality of the English language, just minor things like line 32 "An influenza epidemic were expected ...". Things that are very easily to correct as the manuscript is highly readable. 

Reviewer 3 Report

I have reviewed this informative article. The quality of the abstract needs improvement.

This study describes that main aim of this study was to compare impressions of COVID-19 vaccination and influenza vaccination in Japan by analyzing social media (Twitter) using a text-mining method. We obtained 10,000 tweets using the keywords “corona vaccine” and “influenza vaccine” on December 15, 2022 and February 19, 2023. We then counted the number of times the words were used and listed frequently of these words by a text-mining method called KH Coder. We also investigated concepts in the data using groups of words that often appeared together or groups of documents that contained the same words using multi-dimensional scaling (MDS). “Death” in relation to corona vaccine and “severe disease” for influenza vaccine were frequently used on December 15, 2022. The number of times the word “death” was used decreased, “after effect” was newly recognized for corona vaccine, and “severe disease” was not used in relation to influenza vaccine. By analyzing a large sample of tweets from Twitter, a marked difference was observed in impressions of corona vaccination and influenza vaccination in Japan.

Please revise the article and remove minor English grammar errors. I suggest the authors take English editing services from some agencies to improve the quality of this study. I am suggesting some studies. Please read these studies and improve your article.

Introduction section

I suggest that authors to read the suggested studies and add the latest citations to the introduction, literature and method sections to enhance the quality of the study.v

Su, Z., Cheshmehzangi, A., Bentley, B. L., McDonnell, D., Segalo, S., Ahmad, J., . . . da Veiga, C. P. (2022). Technology-based interventions for health challenges older women face amid COVID-19: a systematic review protocol. Syst Rev, 11(1), 271. doi:10.1186/s13643-022-02150-9

Abbas, J. (2021). Crisis management, transnational healthcare challenges and opportunities: The intersection of COVID-19 pandemic and global mental health. Research in Globalization, 3, 100037. doi:10.1016/j.resglo.2021.100037

Literature section:

Add literature section. You cannot delete this section. Read the suggested literature studies to enhance your work's quality. Add a few lines about studies on how education and social media can educate people.

Abbas, J., Al-Sulaiti, K., Lorente, D. B., Shah, S. A. R., & Shahzad, U. (2022). Reset the industry redux through corporate social responsibility: The COVID-19 tourism impact on hospitality firms through business model innovation. In Economic Growth and Environmental Quality in a Post-Pandemic World (1st` ed., pp. 177-201): Routledge.

Micah, A. E., Bhangdia, K., Cogswell, I. E., Lasher, D., Lidral-Porter, B., Maddison, E. R., . . . Dieleman, J. L. (2023). Global investments in pandemic preparedness and COVID-19: development assistance and domestic spending on health between 1990 and 2026. The Lancet Global Health, 11(3), e385-e413. doi:10.1016/S2214-109X(23)00007-4

Farzadfar, F., Naghavi, M., Sepanlou, S. G., Saeedi Moghaddam, S., Dangel, W. J., Davis Weaver, N., . . . Larijani, B. (2022). Health system performance in Iran: a systematic analysis for the Global Burden of Disease Study 2019. The Lancet, 399(10335), 1625-1645. doi:10.1016/S0140-6736(21)02751-3

Materials and Methods

This section is very weak. Please follow the suggested studies and improve your paper. The authors need to improve this section. I am recommending some good studies. Read the methods of these studies, and improve your paper. Suggested useful articles citations:

Hafeez, A., Dangel, W. J., Ostroff, S. M., Kiani, A. G., Glenn, S. D., . . . Mokdad, A. H. (2023). The state of health in Pakistan and its provinces and territories, 1990–2019: a systematic analysis for the Global Burden of Disease Study 2019. The Lancet Global Health, 11(2), e229-e243. doi:https://doi.org/10.1016/S2214-109X(22)00497-1

Micah, A. E., Bhangdia, K., Cogswell, I. E., Lasher, D., Lidral-Porter, B., Maddison, E. R., . . . Dieleman, J. L. (2023). Global investments in pandemic preparedness and COVID-19: development assistance and domestic spending on health between 1990 and 2026. The Lancet Global Health. doi:https://doi.org/10.1016/S2214-109X(23)00007-4

Result

Read the results of these studies, and improve your paper according to these studies in this section. Suggested useful articles citations

Discussion section:

The separate heading of the discussion section should be around one page. Improve the study and make it strong. See the recommended studies and improve your sections.

Conclusion

Highpoint creativity and scientific contribution of this study to the body of literature. The English level needs corrections to meet scientific merit for publication. I accept and endorse this manuscript for publication after minor corrections, as suggested.

Moderate editing of the English language is required.

Reviewer 4 Report

Dear Authors,

Please take into your consideration the following issues.

Please add more information about the way that social media and in particular Twitter transmit intentions, opinions, and ideas regarding COVID-19 since this is the aim of your study. Make clear the way that Twitter could have an association with COVID-19 vaccination campaigns.

Please, add a paragraph about the public health impact of your study results.

Please use only the abbreviation COVID-19 in the title.

Please, add percentages in Tables 1 and 2.

Round 2

Reviewer 2 Report

I apprecite the effort that the authors have dedicated to improving the manuscript by following most of my reccomendations.

I suggest that the manuscript is almost ready for publication but there are little things that do need to be sorted out. Just as an example in the acknowledgement it is written that "... we grateful thank for helping ediging..." (line 371). I urge the authors to read very carefully through the manuscript before submitting the final version as the research is interesting and deserves a little more attention to detail.

I kindly ask the authors to read carefully through the manuscript before submitting the final version. This research is very interesting but there are some little things that actually might distract the attention of any interested reader away from the content and the overall arguments that the authors are making.
